# Risk Factors Associated with Diarrheal Episodes in an Agricultural Community in Nam Dinh Province, Vietnam: A Prospective Cohort Study

**DOI:** 10.3390/ijerph19042456

**Published:** 2022-02-21

**Authors:** Hanako Iwashita, Asako Tokizawa, Vu Dinh Thiem, Taichiro Takemura, Tuan Hai Nguyen, Hang Thi Doan, Anh Hong Quynh Pham, Na Ly Tran, Tetsu Yamashiro

**Affiliations:** 1Department of International Affairs and Tropical Medicine, Tokyo Women’s Medical University, Tokyo 162-8666, Japan; 2Research Center for Child Mental Development, Hamamatsu University School of Medicine, Shizuoka 431-3192, Japan; tokizawa@hama-med.ac.jp; 3National Institute of Hygiene and Epidemiology, Hanoi 100000, Vietnam; vdt@nihe.org.vn (V.D.T.); nhaituan@yahoo.com (T.H.N.); 4Vietnam Research Station, Center for Infectious Disease Research in Asia and Africa, Institute of Tropical Medicine, Nagasaki University, Nagasaki 852-8523, Japan; taichiro@nagasaki-u.ac.jp; 5VITECH DEVELOPMENT CO., Ltd., Hanoi 700000, Vietnam; doanhang.vitech@gmail.com; 6Department of Biomedical Chemistry, Graduate of Medicine, The University of Tokyo, Tokyo 113-8654, Japan; phamhongquynhanh.k56@hus.edu.vn; 7Division of Bio-Medical Science & Technology, Korea University of Science and Technology (UST), Daejeon 34113, Korea; lynatran.14@gmail.com; 8Department of Bacteriology, Graduate School of Medicine, University of the Ryukyus, Okinawa 903-0215, Japan

**Keywords:** diarrhea, risk factor, drinking water, toilet, sanitation, infrastructure, prospective study, cohort study, community, Vietnam

## Abstract

In Vietnam, data on the risk factors for diarrhea at the community level remain sparse. This study aimed to provide an overview of diarrheal diseases in an agricultural community in Vietnam, targeting all age groups. Specifically, we investigated the incidence of diarrheal disease at the community level and described the potential risk factors associated with diarrheal diseases. In this prospective cohort study, a total of 1508 residents were enrolled during the 54-week study period in northern Vietnam. The observed diarrheal incidence per person-year was 0.51 episodes. For children aged <5 years, the incidence per person-year was 0.81 episodes. Unexpectedly, the frequency of diarrhea was significantly higher among participants who used tap water for drinking than among participants who used rainwater. Participants who used a flush toilet had less frequent diarrhea than those who used a pit latrine. The potential risk factors for diarrhea included the source of water used in daily life, drinking water, and type of toilet. However, the direct reason for the association between potential risk factors and diarrhea was not clear. The infection routes of diarrheal pathogens in the environment remain to be investigated at this study site.

## 1. Introduction

Diarrheal diseases are a significant issue for global health. Worldwide, of the nearly 5.3 million deaths of children aged less than 5 years in 2018, 8% were attributed to diarrhea, arising from a combination of factors [1], including poverty [2,3], malnutrition [4], poor sanitation and hygiene [5], unsafe drinking water [6,7], and poor healthcare systems [8]. In recent years, the mortality associated with diarrhea among children living in developed countries has shown a trend of rapid decline [9]. Although Vietnam has a rapidly developing economy, diarrheal disease remains the leading cause of morbidity and mortality among children aged less than 5 years [10].

Hospital-based studies, rather than community-based studies, have been conducted preferentially in Vietnam [11,12,13,14,15,16], because the method is simpler and less expensive, and involves the handling of severe hospitalized cases having higher priority. A community-based study, however, may establish the actual incidence of diarrhea at the community level. Our study targeted all age groups and examined repeated diarrhea cases in the same individual within each household. It is believed this will contribute to the understanding of the risk factors for diarrhea. It may also lead to methods of community-based management to prevent diarrhea. Using a variety of approaches and study designs can provide insights into the etiology of diarrhea.

In Vietnam, there have been a variety of reports of transmission pathways at the local level that can lead to the transmission of diarrhea [17]. For example, in rural Vietnam, proximity to livestock, which is vital for households’ subsistence, may provide an opportunity for the transmission of diarrheal pathogens from livestock to humans [17,18]. The risks of diarrhea are not limited to contact with animal feces. Numerous other risk factors for diarrhea should be considered, such as socioeconomic status, infrastructure and sanitation (e.g., water supply, availability of drinking water), and household living conditions (e.g., number of children aged less than 5 years in the household, type of toilet used) [19]. Therefore, by investigating these factors related to socioeconomic and environmental issues, we hoped to uncover hidden factors contributing to diarrheal incidence at our rural Vietnamese study site.

The aim of this study was to estimate the incidence of diarrhea among residents in all age groups in the Hien Khanh commune, Nam Dinh province, northern Vietnam, and its possible risk factors. To the best of our knowledge, this is the first community-based, prospective, longitudinal study exploring the incidence and risk factors for diarrhea in this area.

## 2. Materials and Methods

### 2.1. Study Site and Population

A community-based, prospective cohort study was conducted among a rural population in the Hien Khanh commune, Vu Ban district, Nam Dinh province, northern Vietnam. The “commune” represents a core unit for implementing public health administrative services; hence, the medical information from the participating residents was collected in the study. The Hien Khanh commune comprises 2133 households, with a total population of 7966 people. The community is made up of one ethnic group, Kin. This commune is situated in a rural setting in Nam Dinh Province in the Red River Delta region of northern Vietnam, covering 12 km^2^ and 15 m above sea level. The commune lies 10 km from Nam Dinh city, and approximately 70 km southeast from the capital city, Hanoi. The livelihoods of most people in the region rely on agricultural activities. Notably, tap water was accessible from a single water purification plant (WPP) in the study site. There was no public sewage system.

### 2.2. Research Team

Our research team consisted of a medical doctor and health workers who work at the Hien Khanh commune health center. They have been trained and working as health service staff in the Hien Khanh commune for several years. Thirteen health workers were recruited to participate in this study.

### 2.3. Participant Inclusion

The participant inclusion was based on information provided by the Provincial Preventive Medical Center of the Hien Khanh commune. The enrolment criteria included households with children aged less than 5 years and those households who agreed to participate in the study. There were 435 households with children aged less than 5 years, accounting for 20.4% of all the households in the commune. Of those, 311 households with 1662 individuals that met the criteria were chosen and enrolled; then, 154 individuals were excluded from the study due to moving out of the commune during the study period. Births and any deaths of participants during the study period were not included in this analysis. In total, 1508 participants completed the whole 54-week follow-up, from 27 October 2014 to 16 November 2015 (Figure 1).

### 2.4. Baseline Information through Face-to-Face Interviews Using a Questionnaire

In September 2014, trained health workers began to visit designated candidate households. The enrolment criteria for participation in this study were once again confirmed with the head of each household or with individual household members. If the household met the criteria, the health workers collected the baseline information of each household through face-to-face interviews and using a questionnaire. This questionnaire collected information about the demographic and socioeconomic characteristics of the household, including age; sex; the number of household members; water sources for daily use, and water for drinking (e.g., tap water, water truck, tube well/hand pump, open well, rainwater, canal/river, lake/pond, others); water treatment (e.g., boiling water before drinking or not); distance from the WPP; type of the toilet (e.g., flush toilet with septic tank, pit latrine); livestock animal husbandry (e.g., pigs, buffalos, dogs, cattle, cats, chickens, ducks, or geese); and wealth index (Appendix A). Multiple-choice questions were used to inquire about the water source used for daily use. A representative from each household was provided with multiple options. Based on the responses, each water source was categorized as “used” or “not used” by each household. Toilet facility types were categorized into a flush toilet with septic tank, pit latrine, or other (not categorized).

Household crowding and lower household socioeconomic status could be risk factors for diarrhea [20,21,22]. To evaluate household crowding, we calculated housing space (m^2^) per person as an index of crowding, with a cut-off value of 12 m^2^, based on guidelines for healthy housing produced by the World Health Organization (WHO) [23]. To assess the socioeconomic status of each household, data on the ownership of assets were collected using the principal component analysis (PCA) methodology [24,25,26]. The respondents were apportioned into three levels of wealth considering the possession of household assets, including radio, refrigerator, television, video cassette recorder (VCR), motorcycle, bicycle, car, mobile phone, landline phone, fan, washing machine, sewing machine, air conditioner, computer, Internet, and rice mill machine.

Questionnaire forms were translated from English to Vietnamese and then back-translated to ensure accuracy when completing the final version. Any discrepancies were resolved by discussion among the research team.

### 2.5. Global Positioning System (GPS) for Measuring Distances

The GPS coordinates of all houses, the health center, and the WPP were collected, and the distance from each household to the WPP was calculated using QGIS software (version 2.8.2, Open-source software, Raleigh, NC, USA).

### 2.6. Definition of Diarrhea

We defined diarrhea as the presence of three or more watery or loose fecal discharges within 24 h, following WHO criteria [27]. Any reoccurrence of diarrhea within 14 days was considered part of the prior diarrheal episode, whereas an episode of diarrhea after a diarrhea-free gap of 14 or more days was taken to represent a new episode [28].

### 2.7. Routine Follow-Up for Collecting Data for Diarrhea

This study was carried out for 54 weeks, from 27 October 2014 to 16 November 2015. For longitudinal diarrheal surveillance, each health worker arranged for twice-weekly visits to between 15 and 60 households, to ask whether any member of the household had reported diarrheal episodes. If they reported an episode of diarrhea, a health worker received the diarrheal sample in a sample pot. Meanwhile, data on the occurrence of diarrhea were also collected through interviews with those participants who had suffered with diarrhea or, in case of children, their family members. Sample pots were distributed to all participating households beforehand. When the cups ran out, they were immediately redistributed to households. Diarrheal samples were analyzed in a different study. In the current study, the diarrheal episode was counted as such when accompanied by the diarrheal sample. The confirmed data on diarrhea were used for the analysis. A meeting was held once per month at the Hien Khanh commune health station to ensure that all procedures were undertaken properly by the health workers. Data collection was ceased for the national new year Tet holidays for two weeks in February 2015.

### 2.8. Data Management

Data management was performed at the Hien Khanh commune health station and the National Institute of Hygiene and Epidemiology. Data were double-entered and managed using FoxPro 7.0 software (Microsoft, Lewistown, PA, USA).

### 2.9. Risk Factor Analysis

We employed maximum likelihood statistical models to assess the association between various risk factors, including basic demographic factors and diarrheal episodes. All statistical analyses were carried out with R version 3.5.0 (R Core Team, Vienna, Austria), using the library glmmADMB package [29]. As the number of diarrheal episodes is over-dispersed, a negative binomial generalized linear mixed model (NB-GLMM) was employed. The response variable was the number of diarrhea episodes. The explanatory variables were demographic and environmental factors. Demographic factors included age, sex, and level of wealth. Other factors included the use or non-use of each source of water for daily use (tap water, water truck, tube well, open well, rainwater, or lake/pond); the source of drinking water (tap water, rainwater, or other); the distance from the WPP; the type of toilet facility (flush toilet with septic tank, pit latrine, or other); and the presence or absence of animal husbandry (pigs, buffalos, dogs, cattle, cats, chickens, ducks, geese, or other animals). Behavioral factors, such as the practice of boiling water, were also included. As each household comprised several residents, “household” was considered a random factor and was added [30]. First, to perform the univariate NB-GLMM, each explanatory variable was separately entered into the model, with “household” as a random factor. Second, to control for confounding variables, a multivariate NB-GLMM was performed. The response and explanatory variables used were the same as those in the univariate NB-GLMM and were entered into the multivariate NB-GLMM with “household” again as a random factor. If explanatory variables showed variance inflation factor (VIF) values of more than 5, there would likely be a multicollinearity issue [30]. We would then remove one or more variables and re-evaluate the potential new model variable configuration. We repeated this analysis until all explanatory variables had VIF values of less than 5 and suitable combinations of variables had been determined. In each case of multivariate NB-GLMM with relevant variables, the best model for predicting the factors affecting episodes of diarrhea was chosen using a backward stepwise model selection, based on the Akaike Information Criterion (AIC) [31]. Confidence intervals of 95% and *p* values of less than 0.05 were considered to be statistically significant. For all analyses of local risk factors for diarrheal episodes, we used participants’ age when the study began (27 October 2014).

## 3. Results

### 3.1. Socioeconomic Characteristics of Participants

The study population consisted of 311 households, with a total of 1508 residents who were aged from 0 to 97 years. The geographical distribution of the randomly selected households is shown in the map (Figure 2). Throughout this portion of the study, between the end of October 2014 and the middle of November 2015, a total of 311 households completed follow-up. We did not use any data from residents of the households who were born or died during our study period. The demographic characteristics of the participants are summarized in Table 1. More than half of the participants were female (56.4%). The mean number of household members was 4.86 (SD: 1.22). The mean number of children aged less than 5 years in each household was 1.28 (SD: 0.50). All 311 households lived in houses made of similar materials, i.e., bricks. The mean housing area was 75.1 m^2^ (SD: 35.6 m^2^). The proportion of households who used tap water for daily use was 92.6%, whereas 65.3% used rainwater. No households used canal or river water for daily use; therefore, we excluded it as a variable for analysis. Although one household answered “Other” about the water source used for daily use, the details of this other water source were not specified, and this risk-factor variable was not used for the analysis. Concerning the variables of water for drinking, we categorized these into three types of water sources: tap water, rainwater, and others. The proportion of households who used tap water, rainwater, and others for drinking was 35.1%, 64.4%, and 0.5%, respectively. The answer of “Other” for drinking water included mineral water (four individuals) and truck water (three individuals), which we combined into one category, “Other”, for the analysis. For the type of toilet facility, 78.4% of households had a flush toilet (with septic tank), 17.8% used a pit latrine, and the remaining 3.7% were of other types. The distribution of distances from the WPP showed two peaks and was categorized into “near” (<1.5 km from the WPP) and “far” (≥1.5 km from the WPP) (Appendix A). The education level achieved among individuals aged more than 15 years was as follows: lower than primary school, 28/907 (3.1%); primary school, 52/907 (5.7%); secondary high-school, 475/907 (52.4%); high school, 346/907 (38.1%); intermediate college, 5/907 (5.5%); and university, 1/907 (0.1%). These data were only used for basic demographic information and not for any further analysis because of the nature of this study design, which was focused on all age groups, including children who were yet to receive any formal education. Most households (90.7%) owned animals; 13.7% had pigs, 1.9% had buffalos, 73.9% had dogs, 10.7% had cattle, 52.1% had cats, 70.9% had chickens, and 17.4% had ducks or geese. In this study site, wealth disparities were not that large, based on the data relating to the ownership of household assets (Appendix A).

### 3.2. Diarrheal Episodes

We identified 572/1508 participants who reported at least one episode of diarrhea during the study period. In total, there were 791 episodes of diarrhea over 1561.7 person-years (study period × total number of participants: 1.04 years (54 weeks) × 1508 participants) in the study population. The cumulative number of diarrheal episodes per individual during the study period varied widely: 936/1508 (62.1%) had none; 407/1508 (27.0%) had one episode; 125/1508 (8.3%) had two episodes; 32/1508 (2.1%) had three episodes; 5/1508 (0.3%) had four episodes; 2/1508 (0.1%) had five episodes; and 1/1508 (0.1%) had eight episodes. The case with the highest number of episodes was that of a 2-year-old child. The population age distribution and the individuals who had at least one episode of diarrhea are shown in Figure 3. The age histogram shows three peaks, at around 5, 30, and 60 years of age (Figure 3). Due to the nature of the study, we focused on households with children aged less than 5 years. Individuals with diarrhea (colored in Figure 3) showed three similar peaks. Overall, the incidence of diarrheal episodes was 0.51 episodes per person-year (95% CI 0.47–0.55). For children aged less than 5 years, the incidence was 0.81 episodes per person-year (95% CI 0.71–0.91). By breaking this group down into smaller groups of one year old each, each separated by one year of age, the results are as follows: under 1 year, 1.04 episodes per person-year (95% CI 0.67–1.41); 1–2 years, 0.86 (0.67–1.06); 2–3 years, 0.83 (0.61–1.06); 3–4 years, 0.82 (0.62–1.01); 4–5 years, 0.60 (0.40–0.82). We found that the younger the child, the higher the incidence (Appendix A, Appendix A). There was no difference in the occurrence of diarrhea between males and females.

### 3.3. Diarrheal Risk Factors

Based on the univariate NB-GLMM, age, rainwater for daily use, drinking water (tap water versus rainwater), distance from the WPP, and toilet facility type (flush toilet versus pit latrine) were significantly associated with diarrheal episodes. Participants who used rainwater for daily use showed a significantly lower risk of diarrhea than those who did not (incidence rate ratio: IRR 0.67, 95% CI 0.53–0.85). For participants who used tap water for drinking, the risk of diarrhea was significantly higher than those who used rainwater (IRR 1.54, 95% CI 1.21–1.95) (Table 2). More than 99% (1500/1508) of the participants boiled the water before drinking and there was no significant difference in the association with diarrhea in the study. Regarding domestic animal husbandry, no significant difference was found between participants who kept animals and those who did not. Ownership of household livestock of any kind did not affect the incidence of diarrhea (Table 2). In terms of toilet facility type, a significant trend of reduced risk was found among participants using a flush toilet (with septic tank) versus those using a pit latrine (IRR 0.69, 95% CI 0.51–0.92). Among participants who lived near (<1.5 km) the WPP, a higher number of diarrheal episodes were reported versus participants who lived far (≥1.5 km) from the WPP (IRR 1.43, 95% CI 1.1–1.86) (Table 2).

The explanatory variables that had a significant relationship with the number of diarrheal episodes in the multivariate NB-GLMM, such as age, rainwater for daily use, water for drinking, distance from the WPP, and type of toilet facility, were also significant in the univariate NB-GLMM (Table 2 and Table 3). Regarding the model selection, a complete model with all explanatory variables for multivariate NB-GLMM was initially prepared. Then, the explanatory variable rainwater for daily use was excluded from the model because the variables rainwater for daily use and water for drinking had high collinearity (VIF > 6) (Table 3). After removing the explanatory variable, water for drinking, from the model, multicollinearity was no longer present for any of the other explanatory variables (all VIF < 3), and the model selection was finalized. As a result, the final model retained five explanatory variables—age, tap water for daily use, rainwater for daily use, distance from the WPP, and toilet facility type—that were significant (Table 3 (Model 1), Appendix A). Additionally, we selected another model with explanatory variables similar to the previous multivariate NB-GLMM, except that rainwater for daily use was excluded and water for drinking was included instead (Table 3 (Model 2)). The results did not change compared with the previous results, except that the water for the drinking variable was retained rather than the rainwater for daily use variable (Appendix A). The difference in the AIC between the two models was less than 1, indicating equally supported models. In terms of behavior related to the use of tap water for daily use, multivariate analysis showed a significantly higher number of diarrheal episodes in participants who used tap water for daily use versus those who did not use tap water for daily use (IRR 1.90, 95% CI 1.16–3.09). On the other hand, univariate analysis showed a higher risk, but this was not significant (IRR 1.59, 95% CI 0.97–2.62). This result was due to no adjustment having been made for multiplicity, which was applied for the multiple testing due to the exploratory nature of the analysis. There were no significant discrepancies between the results of the univariate NB-GLMM and multivariate NB-GLMM, except for the differences in significance, as described above. All results reflecting the increased risk for diarrheal episodes related to both tap water for daily use and water for drinking were essentially similar.

## 4. Discussion

This report presents an assessment of the incidence of diarrheal episodes among all age groups in our cohort study and the characteristics of participants concerning the frequency of diarrheal episodes.

### 4.1. Incidence of Recurrent Diarrhea

In our study, the incidence of diarrhea in all age groups was 0.51 episodes/person-year (95% CI 0.47–0.55), whereas the incidence in children aged less than 5 years only was 0.81 episodes/person-year (95% CI 0.71–0.91). Many studies have investigated diarrhea in children aged less than 5 years [28,32,33,34], which is the age group most vulnerable for diarrheal morbidity and mortality. However, our study targeted all age groups because we were interested in community-based management of diarrhea, including communicable types of diarrhea caused by, e.g., norovirus. Generally speaking, cohort studies targeting all age groups tend to show lower incidence rates than those targeting only children [35], although the results may differ slightly depending on the study design and area.

A systematic review by Fischer and colleagues identified community-based cohort studies of children aged 0 to 59 months in developing countries [36]. They reported that the estimated incidence of diarrhea in children under 5 declined, from 3.4 episodes per person-year in 1990 to 2.9 episodes per person-year in 2010 [36]. Although we could not find the actual incidence in 2015 using the methodology adopted in the present study, we assume that the incidence may still be higher than our data in 2015, which was 0.84 episodes per person-year. Interpretations of incidence are difficult to compare due to the context-specific situation of each study area, although our study also reported a similar trend of higher rates of diarrheal incidence in children aged less than 5 years.

Some studies have been conducted by other research teams in Vietnam that focused only on adults [35,37] or all age groups combined [38]. For example, a study conducted by Pham-Duc et al. in Hanan province of northern Vietnam, an agricultural area with a similar climate and environmental conditions to those at our study site, reported a diarrheal incidence rate of 0.28 episodes per person-year in an open cohort of 867 adults aged 16 to 65 years [37]. Their report did not include any children aged less than 15 years. In a longitudinal study that included all age groups, in both urban and rural areas, in India, an overall incidence rate among all age groups of 0.12 (95% CI 0.11–0.14) episodes/person-year was reported, whereas the rate among children aged less than 5 years was 0.51 (95% CI 0.44–0.58) episodes/person-year [39]. Among children aged less than 5 years, the incidence of diarrhea was highest in infants (aged < 1 year) at 1.07 episodes/person-year [39]. Individuals living in rural areas exhibited a lower incidence rate of diarrhea.

### 4.2. Diarrheal Risk Factors

Contrary to expectations, we found a trend of higher rates of diarrheal disease in households that used tap water rather than rainwater. There are various reports on the relationship between diarrhea and tap water or rainwater. A study in Kabul, Afghanistan, reported that approximately half of all households in five districts within Kabul used home piped water [32]. Here, the authors reported that a tendency of risk reduction was seen among households using an open well versus a piped water source, and they concluded that no association between disease prevention and piped water was found. A cross-sectional study conducted in a second city in Senegal also found no significant association between the source of drinking water and the occurrence of diarrhea [40].

Conversely, a cohort study in southern Vietnam reported that a lack of access to tap water was significantly associated with increased hospitalization due to diarrhea [16]. Similarly, Pham-Duc et al. reported the benefits of tap water in their nested case–control study in an agricultural community in northern Vietnam [37]. Based on a conditional logistic regression analysis, their data showed that participants who lived in households using rainwater for drinking had a significantly higher risk of diarrhea than those living in households using tap water for drinking [37]. Therefore, our findings were not consistent with those of the studies conducted in other parts of Vietnam, which demonstrated the benefits of drinking tap water for reducing the incidence of diarrhea.

Although studies often report that public water supply systems can reduce diarrheal burden [41], the systems themselves may sometimes generate their dangers due to the complexity of continued management of the entire supply system. For instance, maintenance of the water supply system is difficult in our study area, where power failures frequently occur [42]. Therefore, transient changes in water pressure caused by power failures can influence water quality [43]. This might be expected to be stronger for households closer to WPP. LeChevallier et al. reported methods to prevent the intrusion of contaminants that may cause health problems. The American Water Works Association Research Foundation (AWWARF) report showed that having low water pressure in otherwise satisfactory water distribution pipes can induce the aspiration into pipes of enteric organisms present in the soil surrounding the pipes [43]. If water pipes burst, it can negate any health benefits if the water is contaminated, although there were no records of this type of incident in our study area. Even in the absence of an actual burst pipe, however, low water pressure in distribution systems and intermittent supply are notorious risk factors for outbreaks of waterborne disease [43].

In our study site, rooftop rainwater harvesting is one of the main water sources; this practice began a few decades ago before tap water was piped into this area. In 2011, Ozdemir et al. reported that rainwater harvesting at the household level was widely practiced and was a primary source of drinking water in Southeast Asia; it was also economically feasible in southern Vietnam [44]. However, Meera et al. had already noted in 2006 that there was considerable contamination of household-harvested rainwater with pathogenic microorganisms [45]. Other studies conducted in Australia and the U.K. also showed the presence of a large variety of pathogenic microorganisms in household-harvested rainwater [46,47]. Hamilton et al. summarized the benefits and harm rainwater can present for human health [48]. Their review showed that microbial contamination of rainwater had been identified in many epidemiological studies worldwide, but it is possible to develop contamination prevention strategies using a variety of approaches.

Even if the water source is slightly contaminated, there are ways to ensure safety at the point of use. One of them is boiling water, which is used to make tea in Vietnamese social customs. Anecdotal reports from residents in our study site suggest that rainwater was considered ideal for sweetening tea. Conversely, they tended to be hesitant to drink tap water for tea because of the smell of chlorine [49]. Our questionnaire survey showed that boiling was a common answer given as a treatment practice. It is likely to help maximize the safety of rainwater, especially when consumed for tea. However, we do not know precisely how long water was boiled prior to drinking or which water was boiled.

Based on the limited information obtained from this study, it can be strongly recommended that the toilet facilities be improved to the flush toilet type to prevent diarrhea. Other studies have also compared the incidence of diarrhea in flush toilets and pit latrines. A historic cohort study conducted in Kenya with HIV-infected mothers and their infants showed an association between flush toilets and a reduced risk of diarrhea [50]. Improved toilets reduce the risks of both diarrhea and stunting in children [51]. Designing toilets in a way that facilitates hygienic behavior may help prevent diarrhea. As in the study conducted in Laos, providing soap at handwashing facilities and making it available may help reduce the risk of diarrhea [52]. In order to reduce the burden of diarrheal diseases more effectively, community-wide sanitation can be introduced in addition to household-level sanitation [53].

### 4.3. Potential Risk Factors That Remain Unidentified Following Our Study

There are further concerns regarding potential risk factors that remain unidentified following our study. For instance, most households were unlikely to be aware of the duration for which their rainwater was stored, as it is continuously harvested from cumulative rainfall, calling into question the extent of the presence of pathogens. The use of tap water following a power failure may also be a risk factor for diarrhea [54]. Respondents could not clearly remember when their tap water supply had stopped and restarted due to frequent power failures [42]. Targeted research into point-of-use water may facilitate the development of effective water storage systems at a household level to effectively protect against these dangers.

The presence of animals near human dwellings has been reported to be associated with diarrhea in humans [55], although our study showed no increased risk for diarrhea linked with animal husbandry. In cases of diarrhea caused by *Giardia* spp. and *Cryptosporidium* spp., transmission between animals and humans was suspected due to environmental contamination with both animal and human feces in the same setting [18,56]. Fortunately, this transmission did not increase the risk of diarrhea. Although other studies have also considered open-field defecation in their analysis, we did not, because all households had toilet facilities.

### 4.4. Limitations

Our study has several limitations. First, there was no way to authenticate the actual occurrence of diarrhea using our passive diarrhea detection system. Although all participants in our study had already been registered at the Hien Khanh commune health station, they could choose to consult their preferred commune health stations without relying on the system they were registered with [57]. This could have led to lower estimations of the incidence of diarrhea. However, in our study site, participants seemed willing to consult the Hien Khanh commune health station to address any diarrheal disease. Participants tended not to use other commune health stations in cases of diarrhea because the Hien Khanh commune health station was easily accessible and more convenient. Moreover, trained health workers who were staff members at the Hien Khanh commune health station were hired for this study, then trained to communicate with both participants and physicians and correctly define diarrhea according to WHO criteria [58]. This system reduced the likelihood of missing cases of diarrhea.

Second, in this study, only the incidence of diarrheal episodes was used as a response variable to analyze the relationship between risk factors and household socioeconomic and environmental characteristics. Ideally, a more comprehensive approach, using more precise response or explanatory variables, such as the presence of diarrheal pathogens in human fecal samples and/or water samples, would have been used to confirm diarrheal pathogen transmission pathways. Although many types of pathogens cause diarrhea, with each pathogen manifesting its complex, dynamic transmission patterns, pathogen-specific epidemiological data would have been useful for making assumptions about diarrheal pathogen exposure and the eventual development of targeted interventions. To develop appropriate preventive measures, potential contamination sources, particularly water sources at the public and the household levels, must be investigated. If there is a problem with the water pipes, proper mapping information is also important. In this analysis, the distances are straight lines on a map, not the actual lengths of the water pipes. Therefore, at this stage, we can only suggest the importance of factors influencing the incidence of diarrhea, which is still far from developing appropriate prevention measures.

Third, we did not investigate detailed human behaviors other than boiling water. It would have been helpful to investigate the effects of individuals’ practices and attitudes regarding the choice of water source and water treatment. This could differ among individuals and may have affected our study results. However, we could not know who, when, and what kind of practices individuals undertook for which water sources. The lack of detailed information on local treatment practices for water purification just prior to its use is recognized as a limitation of our results. Moreover, our results might have been influenced by a combination of factors, such as the duration of water storage at a household; personal hygiene practices, such as handwashing; and differences in water treatment.

Fourth, although seasonality of diarrheal disease has been recognized in Ho Chi Minh City, southern Vietnam [59], we did not consider factors related to seasonality in our analysis using data from our study area in northern Vietnam, where the climate is humid and tropical (Appendix A). Finally, the severity of diarrheal episodes was not considered. Especially in children, diarrhea can be severe, and it cannot be denied that they may have been more likely to be noticed than adults.

### 4.5. Significance of This Study and Its Implications

Despite these limitations, our study still provides a valuable overview of diarrheal incidence in this agricultural community in northern Vietnam. We were able to identify risk factors, such as water used for daily use and drinking and type of toilet facilities, such as flush toilets or pit latrines. Although we initially assumed that livestock ownership would be one of the main risks for diarrheal disease in our study area, it was found that this was not likely. Our community survey revealed that more targeted studies should be conducted, initially on issues around water sources and the type of toilet. These should include investigating whether and to what extent tap water and rainwater are contaminated, how often power failures occur in this study area, and how power failures can affect water quality. Moreover, if water supply pipes and toilet tanks are in close proximity to each other, it is necessary to verify whether the proximity affects the water quality during leakage.

One of the implications of our study is that it is important to identify households that are not benefiting from sanitation-related infrastructure. In such cases, we recommend basic personal hygiene practices to solve sanitation problems, such as washing hands before eating, boiling drinking water, and storing water properly. Regardless of the origin of water or even if the water was boiled, longer periods of storing water in storage units were shown to be one of the risks for diarrhea [60]. Water quality improvement at the point of water use is suggested to have a more significant effect on preventing diarrhea [61]. There were various point-of-use quality improvement interventions, including boiling and others, such as chlorination, flocculation, filtration, and solar disinfection. Further research may be helpful to estimate the magnitude of the effects of these interventions in different home settings.

## 5. Conclusions

We identified potential risk factors for diarrhea incidence, such as the water source used for daily use and drinking and toilet facility type. Particularly in this rural area in the north of Vietnam, public infrastructure and sanitation are inadequate. Sanitation is a primary barrier to prevent fecal–oral transmission of the pathogens that cause diarrhea. Although pathogen exposure is only one cause of diarrhea, all individuals, especially children, who are most vulnerable to diarrheal morbidity and mortality, should be protected from environmental contamination by diarrheal pathogens. However, our study could not provide sufficient evidence of diarrheal pathogen transmission pathways to suggest concrete preventive measures. Therefore, we encourage further investigation tailored to the sources of potential diarrheal pathogens in this setting. Regardless of the level of development of the infrastructure, it is important to start by improving the rural living conditions at the household level to prevent diarrhea. In particular, domestic treatment systems for drinking water and sanitation of toilets are important in this study area. We are convinced that future research will help guide the development of adapted public health interventions to minimize the burden of diarrheal disease in this particular local community in Vietnam.

## Figures and Tables

**Figure 1 ijerph-19-02456-f001:**
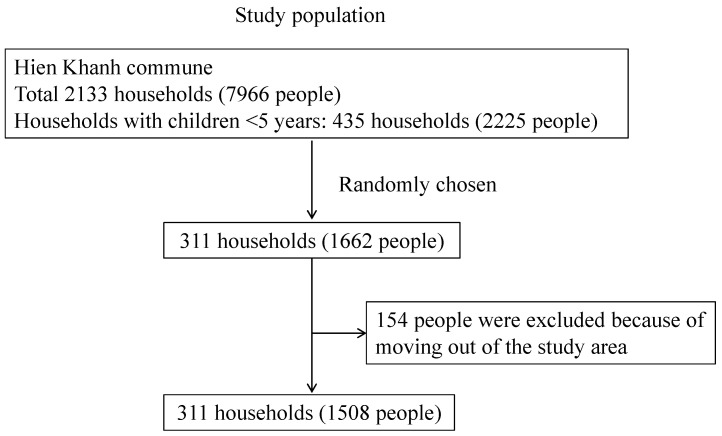
Flow of the participants’ inclusion in the study.

**Figure 2 ijerph-19-02456-f002:**
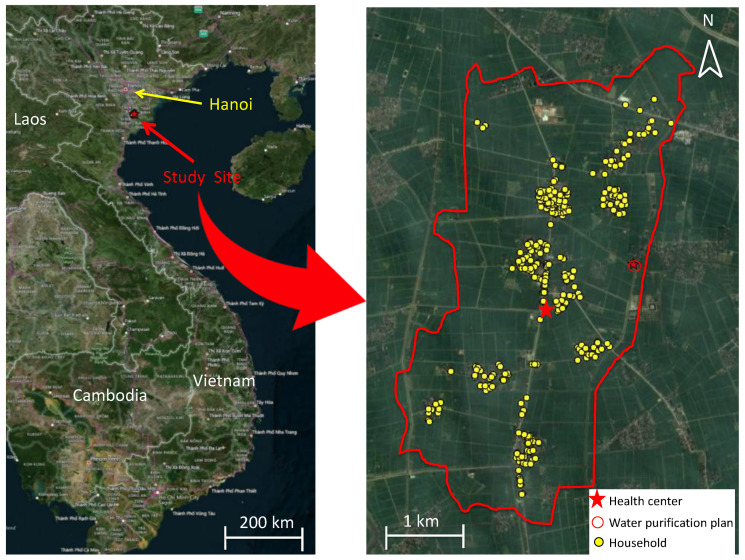
Visualization of sampling locations in Hien Khanh commune.

**Figure 3 ijerph-19-02456-f003:**
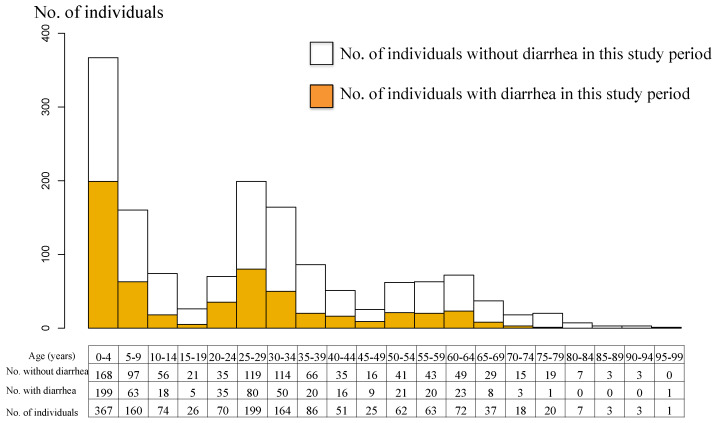
Age pyramid of the study population and diarrhea episode.

**Table 1 ijerph-19-02456-t001:** Characteristics of the household and individuals.

Characteristics	No of Households	No of Individuals	%
Sex			
Male		658	43.6
Female		850	56.4
Age			
0–4		367	24.3
5–14		234	15.5
15–49		621	41.2
50–69		234	15.5
≥70		52	3.4
Number of children under 5 years in the house			
≥2	82		26.4
<2	229		73.6
Household crowding (m^2^/person)			
<12 m^2^	99	536	35.5
≥12 m^2^	212	972	64.5
Water source for daily use			
Tap water for daily use			
Yes	289	1397	92.6
No	22	111	7.4
Water truck for daily use			
Yes	2	8	0.5
No	309	1500	99.5
Tube well/Hand pump for daily use			
Yes	15	84	5.6
No	296	1424	94.4
Open well for daily use			
Yes	6	32	2.1
No	305	1476	97.9
Rainwater for daily use			
Yes	201	985	65.3
No	110	523	34.7
Canal/River for daily use			
Yes	0	0	0.0
No	311	1508	100.0
Lake/Pond for daily use			
Yes	3	16	1.1
No	308	1492	98.9
Others for daily use			
Yes	1	5	0.3
No	310	1503	99.7
Water for drinking			
Tap water	111	530	35.1
Rain water	198	971	64.4
Other (Mineral, Truck)	2	7	0.5
Boiling water			
Yes	309	1500	99.5
No	2	8	0.5
Distance from WWP (water purification plant)			
<1.5 km	217	1056	70.0
≥1.5 km	94	452	30.0
Toilet facility type			
Flush toilet	244	1183	78.4
Pit latrine	55	269	17.8
Others	12	56	3.7
Animals			
Pigs			
Yes	39	206	13.7
No	272	1302	86.3
Buffalos			
Yes	5	28	1.9
No	306	1408	98.1
Dogs			
Yes	215	1114	73.9
No	87	394	26.1
Cattle			
Yes	30	162	10.7
No	281	1346	89.3
Cats			
Yes	158	786	52.1
No	153	722	47.9
Chicken or Birds			
Yes	212	1069	70.9
No	99	439	29.1
Ducks/Geese			
Yes	50	263	17.4
No	261	1245	82.6
Existence of animal			
Any animal	277	1368	90.7
No animal	34	140	9.3
Wealth level			
Low	75	340	22.5
Middle	186	901	59.7
High	50	267	17.7

**Table 2 ijerph-19-02456-t002:** Univariate NB-GLMM of risk factors using the data in this study period (54 weeks). “Household” was included as a random factor.

	Individuals without Diarrhea	Individuals with Diarrhea	Cumulative Episodes of Diarrhea	Person-Years of Observation	Diarrhea Episodes/Person per Year (95% CI)	Incidence Rate Ratio (IRR) (95% CI)	*p* Value
Sex							
Male	423	235	319	681.4	0.47 (0.41–0.52)	1	
Female	513	337	472	880.3	0.54 (0.48–0.59)	1.11 (0.95–1.28)	0.19
Age							
0–4	168	199	283	380.1	0.81 (0.71–0.91)	1.79 (1.52–2.11)	<0.0001 *
5–14	153	81	92	242.3	0.43 (0.34–0.51)	0.99 (0.79–1.26)	0.95
15–49	406	215	308	643.1	0.44 (0.39–0.49)	1	
50–69	162	72	103	242.3	0.38 (0.30–0.46)	0.89 (0.69–1.14)	0.35
≥70	47	5	5	53.9	0.09 (0.01–0.17)	0.24 (0.10–0.59)	0.002 *
Household crowding (m^2^/person)							
<12 m^2^	335	201	278	555.1	0.50 (0.43–0.57)	0.95 (0.73–1.22)	0.66
≥12 m^2^	601	371	513	1006.6	0.51 (0.46–0.56)	1	
Tap water for daily use							
Yes	855	542	750	1446.8	0.52 (0.48–0.56)	1.59 (0.97–2.62)	0.07
No	81	30	41	115.0	0.36 (0.23–0.49)	1	
Water truck for daily use							
Yes	4	4	7	8.3	0.84 (0.05–1.64)	1.77 (0.45–6.98)	0.41
No	932	568	784	1553.4	0.50 (0.47–0.54)	1	
Tube well/Hand pump for daily use							
Yes	41	43	61	87.0	0.70 (0.52–0.88)	1.53 (0.93–2.53)	0.10
No	895	529	730	1474.7	0.50 (0.45–0.54)	1	
Open well for daily use							
Yes	18	14	20	33.1	0.60 (0.31–0.89)	1.17 (0.51–2.66)	0.71
No	918	558	771	1528.6	0.50 (0.46–0.54)	1	
Rainwater for daily use							
Yes	657	328	453	1020.1	0.44 (0.40–0.49)	0.67 (0.53–0.85)	0.001 *
No	279	244	338	541.6	0.62 (0.55–0.69)	1	
Lake/Pond for daily use							
Yes	12	4	8	16.46	0.48 (0.00–1.01)	0.89 (0.26–2.97)	0.84
No	924	568	783	1534.6	0.51 (0.47–0.55)	1	
Water for drinking							
Tap water	279	251	342	548.9	0.62 (0.56–0.69)	1.54 (1.21–1.95)	0.0004 *
Rainwater	656	315	442	1005.6	0.44 (0.39–0.49)	1	
Others (Mineral, Truck)	1	6	7	7.2	0.97 (0.45–1.48)	2.71 (0.73–10.14)	0.14
Boiling water							
Yes	935	565	781	1553.4	0.50 (0.46–0.54)	0.41 (0.13–1.29)	0.13
No	1	7	10	8.3	1.21 (0.64–1.78)	1	
Distance from WPP (water purification plant)							
<1.5 km	624	432	605	1093.6	0.55 (0.50–0.60)	1.43 (1.10–1.86)	0.01 *
≥1.5 km	312	140	186	468.1	0.40 (0.33–0.46)	1	
Toilet facility type							
Flush toilet	765	418	575	1225.1	0.47 (0.43–0.51)	0.69 (0.51–0.92)	0.01 *
Pit latrine	145	124	171	278.6	0.61 (0.52–0.71)	1	
Others	26	30	45	58.0	0.78 (0.51–1.04)	1.21 (0.66–2.22)	0.54
Animal husbandry: Pigs							
Yes	130	76	110	213.3	0.52 (0.40–0.63)	1.04 (0.73–1.48)	0.82
No	806	496	681	1348.4	0.51 (0.46–0.55)	1	
Animal husbandry: Buffalos							
Yes	14	14	16	29.0	0.55 (0.31–0.79)	1.20 (0.50–2.90)	0.69
No	922	558	775	1532.7	0.51 (0.47–0.55)	1	
Animal husbandry: Dogs							
Yes	699	415	568	1153.7	0.49 (0.45–0.54)	0.85 (0.66–1.11)	0.24
No	237	157	223	408.0	0.55 (0.47–0.63)	1	
Animal husbandry: Cattle							
Yes	90	72	89	167.8	0.53 (0.43–0.63)	1.10 (0.75–1.62)	0.63
No	846	500	702	1393.9	0.50 (0.46–0.55)	1	
Animal husbandry: Cats							
Yes	505	281	373	814.0	0.46 (0.41–0.51)	0.86 (0.68–1.08)	0.20
No	431	291	418	747.7	0.56 (0.50–0.62)	1	
Animal husbandry: Chicken or Birds							
Yes	679	390	526	1107.1	0.48 (0.43–0.52)	0.86 (0.66–1.10)	0.23
No	257	182	265	454.6	0.58 (0.50–0.67)	1	
Animal husbandry: Ducks/Geese							
Yes	170	93	121	272.4	0.44 (0.36–0.53)	0.86 (0.63–1.19)	0.38
No	766	479	670	1289.3	0.52 (0.48–0.56)	1	
Animal husbandry: Existence of animals							
Any animal	861	507	704	1416.7	0.50 (0.46–0.54)	1.31 (0.89–1.89)	0.17
No animal	75	65	87	145.0	0.60 (0.48–0.72)	1	
Wealth level							
Low	376	210	305	606.9	0.50 (0.44–0.57)	1	
Middle	261	163	223	439.1	0.51 (0.43–0.58)	1.03 (0.77–1.38)	0.83
High	299	199	263	515.8	0.51 (0.45–0.57)	1.06 (0.80–1.40)	0.71

* Significance at a level of *p* < 0.05.

**Table 3 ijerph-19-02456-t003:** Multivariate NB-GLMM of risk factors using the data in this study period (54 weeks). “Household” was included as a random factor.

	Model 1		Model 2	
	Adjusted Incidence Rate Ratio (IRR) (95% CI)	*p* Value	Adjusted Incidence Rate Ratio (IRR) (95% CI)	*p* Value
Age				
0–4	1.81 (1.54–2.14)	<0.0001 *	1.81 (1.54–2.14)	<0.0001 *
5–14	0.99 (0.78–1.25)	0.90	0.99 (0.78–1.25)	0.92
15–49	1		1	
50–69	0.92 (0.72–1.17)	0.48	0.91 (0.71–1.17)	0.47
≥70	0.24 (0.10–0.59)	0.002 *	0.24 (0.10–0.59)	0.002 *
Water source: Tap water *for daily use*				
Yes	1.90 (1.16–3.09)	0.01 *	1.81 (1.10–2.96)	0.02 *
No	1		1	
Water source: Tube well/Hand pump *for daily use*				
Yes	1.55 (0.98–2.45)	0.06	1.55 (0.98–2.46)	0.06
No	1		1	
Water source: Rainwater *for daily use*				
Yes	0.71 (0.56–0.89)	0.004 *		
No	1			
Water source *for drinking*				
Tap water			1.44 (1.14–1.81)	0.002 *
Rainwater			1	
Others			1.44 (0.42–4.98)	0.57
Boiling water				
Yes	0.41 (0.13–1.29)	0.13	0.41 (0.13–1.29)	0.13
No	1		1	
Distance from the WPP				
<1.5 km	1.62 (1.26–2.08)	0.0002 *	1.62 (1.26–2.08)	0.0002 *
≥1.5 km	1		1	
Toilet facility type				
Flush toilet	0.64 (0.48–0.85)	0.002 *	0.66 (0.49–0.88)	0.004 *
Pit latrine	1		1	
Others	1.05 (0.59–1.85)	0.88	1.12 (0.63–1.98)	0.71
Animal husbandry: Cats				
Yes	0.85 (0.68–1.06)	0.14	0.84 (0.67–1.05)	0.12
No	1		1	
AIC	2684.0		2684.7	

* Significance at a level of *p* < 0.05.

## Data Availability

The data presented in this study are available on request from the corresponding author Hanako Iwashita, at iwashita.hanako@twmu.ac.jp.

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
