# Peer review of "Risk Factors Associated with Diarrheal Episodes in an Agricultural Community in Nam Dinh Province, Vietnam: A Prospective Cohort Study"

_ijerph, 2022, doi:10.3390/ijerph19042456_

Round 1

Reviewer 1 Report

Summary: This manuscript presents results of a 54-week prospective cohort study of 24 hour-recall, self-reported and confirmed diarrhea in a single commune in northern Vietnam, relying on a negative binomial general linear mixed model for estimation of associations between/among explanatory variables and diarrheal disease incidence the response variable (employing both univariate and bivariate models).

Broad reflection: This manuscript is potentially quite informative, presenting a comprehensive portrait of diarrheal illness surveyed bi-weekly in 311 households over the course of a year. In large part, the study is presented clearly and professionally. There are selected questions of clarification that I specify below. My two big concerns relate the reliability of the diarrhea self-reports and “verification” and the suggested causal inference relating to water quality, which was not measured in the study.

Specific comments and questions:

Lines 138-144: First, is it correct that diarrhea episodes are measured via bi-weekly visits by health workers? In other words, is every household visited once every two weeks? The text reads that each health worker visits “5-20” households bi-weekly. 311 HHs split among 13 health workers would mean 24 households per health worker. Second, I am unclear on how “confirmation” of diarrhea episodes occurs. A health worker asks a household for 24 hr recall of diarrhea; if the HH reports the episode, then “a health worker would visit the household and confirm the certainty of the episode by interviewing with the participant him/herself, or with guardians of the child, who reported shed diarrhea.” What does this mean, if the health worker is already at the household doing the survey? Then, “diarrheal samples were supposed to be [sic] collected at each episode by the participants for analyses in a different study.” Were these analyzed for specific pathogens in stool? It is not clear to me that what is being reported here is clinically verified diarrhea. This is important, because significant concerns have been raised about the validity of self-reported diarrhea rather than objective measures of illness. And this confusion is compounded by the acknowledgment at Line 417 “that there is no way to authenticate the actual occurrence of diarrhea.” Please clarify.

Lines 303-332: the discussion of where this study’s findings of diarrheal disease incidence relative to other studies is a bit l. I recommend that the text state clearly and succinctly, “our results are consistent with those reported by other investigators …” or else “our results fall well within the range of those reported by other investigators …

Lines 334-397: the relative risks associated with tap water vs. rain water are indeed notable and unexpected, though collected rainwater being of higher quality than piped water in rural Kenya has been documented in published studies (see Albert et al 2010, doi: 10.1021/es1000566). But lines 352-354, “the systems themselves may sometimes generate their dangers due to the complexity of continued management of the entire supply system” does not make sense as written – some more elaboration on this point would be helpful. Poor management of water supply systems can result in low reliability, and if uptime of the system was low (i.e. disruptions in service were frequent), then households might be forced to source water from much lower quality options. The un-attributed assertion that “transient changes in water pressure caused by power failures” would be more of a problem for households *closer* to the water treatment plant also does not make sense – low pressure would affect those at the ends of the network more acutely. (This raises another concern: is distance from the WPP measured strictly through the GPS positions of the HH and WPP, or does it reflect actual distance of the water supply network? If the former, those GMM results for that explanatory variable would be questionable.) One possible explanation is an interaction between water source (tap vs rainwater vs other) and boiling behavior – that somehow households are more likely to boil harvested rainwater than the other sources (though I realize that the robustness of the boiling benefit is low, at p=0.13). Even more compelling to this reviewer is that there is an unmeasured confounder associated with those relying primarily on rainwater harvesting that is somehow driving diarrheal incidence lower.

Lines 408-410: this finding is an important contribution, especially given the growing recognition of the importance of domestic animal excreta as a reservoir of pathogens transmitted via geophagy by infants and young children. Can you explain why only cats were included in the multivariate model?

Lines 469-470: nothing in the manuscript suggests that households have “difficulty in paying attention to water quality.” Without having measured water quality directly (e.g. E coli or other specific pathogens), nothing can be inferred as to its causal connection to diarrhea.

Author Response

Dear Reviewer 1, 

We would like to thank you for your insightful comments on our paper. 
We would like to respond to you as per the attached file.

Best regards, 

Hanako Iwashita

Reviewer 2 Report

This article provides sufficient background data and statistical analysis, and overall is well-structured. Here are some suggestions, however, for your reference:

  1. Keywords (Lines 34-35): Please use Keywords in a more precise way. Toilet (…), for example, can also be considered a keyword(s)?
  2. Please strengthen the quality of the "introduction".
  3. Suggest redrawing the flowchart (Fig.1) to make it more legible.
  4. May consider adding some environmental photos.
  5. Table 1 , 2 & 3: Please revise. For example: Adjust "All residents" (may put the information in this column elsewhere) to avoid confusion. Also, please distinguish group and sub-group under each characteristic (Suggest not to put them in the same column. Or at least use different fonts for easy identification).

Author Response

Dear Reviewer 2

We wish to express our appreciation for your insightful comment on our paper.
We would like to respond to you as per the attached file.

Best regards, 

Hanako Iwashita

Reviewer 3 Report

This is a very nice manuscript describing a community based prospective cohort study.

There are a few areas where the English language could be edited, this is very minor - it may be helpful to have someone read through it one more time out loud to make sure each sentence flows well.

Table 1 is too big and has too much white space. This information should be condensed to make this table more readable. 

Author Response

Dear Reviewer 3

We wish to express our appreciation for your insightful comment on our paper.
We would like to respond to you as per the attached file.

Best regards, 

Hanako Iwashita

Reviewer 4 Report

This is a very interesting and timely initial study of the risk factors for diarrheal episodes in an agricultural community in Vietnam. The article likely lays the groundwork for other researchers to build on and employ in different contexts to help reveal concrete solutions to prevent diarrheal episodes. The manuscript is ready for publication in the Int. J. Environ. Res. Public Health.

The manuscript is well situated in the existing literature on this important topic, it uses an appropriate research design, acknowledges the limitations of that design and its findings, and is nicely written.

The only minor clarification I would like to see is a mention of cultural aspects that might affect both the incidence of diarrhea and stigma about reporting those cases.  Because the dangers to the health of children under 5 years old are well established and well known, perhaps those cases are more likely to be reported to seek medical help while cases of adults are perhaps not reported and are just dealt with at home.  I also wondered if there is a stigma that prevents reporting of cases, which might depress the incidence rate.

Do people who use tap water treat it differently than those who use "less safe" water sources? Is there a false sense of security?

Lines 480-483 seem out of place and are a little confusing.

Author Response

Dear Reviewer 4

We wish to express our appreciation for your insightful comment on our paper.
We would like to respond to you as per the attached file.

Best regards, 

Hanako Iwashita

Round 2

Reviewer 4 Report

I suggested that this be published earlier, my view remains the same.